# Counting D1-D5-P microstates in supergravity

**Daniel R. Mayerson[1][†] and Masaki Shigemori[2,3][⋆]**

**1** Université Paris-Saclay, CNRS, CEA, Institut de Physique Théorique,
Orme des Merisiers, 91191, Gif-sur-Yvette CEDEX, France
**2** Department of Physics, Nagoya University, Furo-cho, Chikusa-ku, Nagoya 464-8602, Japan
**3** Center for Gravitational Physics, Yukawa Institute for Theoretical Physics, Kyoto University,
Kitashirakawa-Oiwakecho, Sakyo-ku, Kyoto 606-8502, Japan

† daniel.mayerson@ipht.fr, ⋆ masaki.shigemori@nagoya-u.jp

## Abstract

We quantize the D1-D5-P microstate geometries known as superstrata directly in supergravity. We use Rychkov's consistency condition [hep-th/0512053] which was derived for the D1-D5 system; for superstrata, this condition turns out to be strong enough to fix the symplectic form uniquely. For the $(1, 0, n)$ superstrata, we further confirm this quantization by a bona-fide explicit computation of the symplectic form using the semi-classical covariant quantization method in supergravity. We use the resulting quantizations to count the known supergravity superstrata states, finding agreement with previous countings that the number of these states grows parametrically smaller than those of the corresponding black hole.



## Contents



# 1   Introduction and Summary

The fuzzball paradigm in string theory posits that a black hole can be seen as an average geometry over many states with quantum, stringy excitations that extend out to horizon scales [1]. The microstate geometry program aims to explicitly construct as many such microstates as possible as smooth, horizonless solutions in classical supergravity [2, 3]. Such microstate geometries can then be studied within supergravity, providing a unique insight into the microstructure of black hole systems.

The D1-D5 system played an important role as a great success story for the fuzzball paradigm and microstate geometries. The Lunin-Mathur geometries were explicitly constructed [4–6], their CFT duals worked out precisely [7, 8], and finally Rychkov showed that they could be semi-classically quantized in supergravity, reproducing a finite fraction [9] (or all [8, 10]) of the corresponding states as counted in the dual D1-D5 CFT.[1]

However, the D1-D5 system does not correspond to a black hole of finite horizon size, but rather to a geometry where the horizon itself is singular [1]. To obtain a black hole with a finite horizon area, a third charge must be added. The D1-D5-P black hole was the subject of the original holographic counting by Strominger and Vafa [14]; they found that the entropy of this black hole was precisely accounted for by the number of states in the dual CFT with the same quantum numbers. This was also generalized to the BMPV black hole with angular momentum [15].

Although a triumph for black hole physics and holograpy, the Strominger-Vafa counting of D1-D5-P states was done in the dual CFT without any hint towards what these individual states might look like on the supergravity side of the correspondence. This changed with the advent of the *superstrata solutions* [16–19], the smooth, horizonless microstate geometries that each correspond to a single microstate of the D1-D5-P BMPV black hole. There is a large family of known superstrata geometries, although not all superstrata that are believed to exist within this framework have known solutions.[2] Known superstrata solutions are usually parametrized and denoted by three integers $(k, m, n)$; we review their construction briefly in section 2.1.

For these superstrata geometries, the explicit and precise map from supergravity solution to CFT states is known [16, 17, 20, 21]; using this map, one can perform the counting of the superstrata microstates in the CFT [22]. The purpose of this paper is to show that one can also semi-classically quantize the superstrata geometries *directly in supergravity*. This can be done by an application of the same consistency condition that Rychkov used [9] to quantize

---

[1]This may even be considered surprising, as the typical D1-D5 state in supergravity involves structure at scales much smaller than one expects supergravity to be valid [11–13].

[2]Note also that the known superstrata are those that are based on $AdS_3 \times S^3$; more generally one could consider also superstrata on $AdS_3 \times S^3/\mathbb{Z}_k$ or backgrounds with more than one three-cycle.

the D1-D5 microstate geometries in supergravity. As it turns out, this consistency condition is even stronger for superstrata, as it completely fixes the symplectic form — as opposed to the D1-D5 microstates, where it did not fix an overall constant. To give a further support to the correctness of this symplectic form, we also directly quantize the $(1, 0, n)$ superstrata using the semi-classical covariant quantization method in supergravity [23–25]. We show that this family of superstrata can be quantized easily in three dimensions using their recently found dimensional reduction [26], and that this indeed leads to the same symplectic form. It must also be possible to directly quantize the more general $(k, m, n)$ family of superstrata and rederive the symplectic form obtained by Rychkov's consistency condition, although we do not think this would be an interesting exercise. Instead, we emphasize that the consistency condition is quite powerful, in spite of its simplicity, and must have more fruitful applications in the microstate geometry program and elsewhere.

The rest of this paper is structured as follows. In section 2, we give a brief overview of the necessary ingredients of the superstrata geometries we need, and then use Rychkov's consistency condition to find the symplectic form and quantize the (multimode) $(1, 0, n)$ and most general $(k, m, n)$ superstrata. Section 3 contains an explicit, direct verification in supergravity of the $(1, 0, n)$ symplectic form. Then, in section 4, we use the found quantizations to count the $(1, 0, n)$ and general $(k, m, n)$ superstrata geometries; in agreement with earlier counting [22], we find that the number of superstrata geometries grows parametrically smaller than the corresponding black hole entropy. Finally, in appendix A we review the basic elements of symplectic forms, and in appendix B we review Rychkov's original consistency condition argument [9] in the D1-D5 system.

## 2 Superstrata Symplectic Forms

Here, we will use Rychkov's consistency condition (which we review in appendix B) to easily find the symplectic form of superstrata directly in supergravity. First, in section 2.1, we give a brief overview of the necessary ingredients of the general superstrata solutions. Section 2.2 contains a brief overview of the $(1, 0, n)$ subfamily of superstrata solutions, and details how using the Rychkov consistency condition easily leads to the entire symplectic form (19). Then, in section 2.3, we generalize these arguments to find the symplectic form for the most general superstrata in supergravity.

### 2.1 Superstrata overview

We give a brief overview of the superstrata geometries, which are the known microstate geometries for the D1-D5-P black hole. We follow the holomorphic formulation of [18], and give the most important properties of the solution here. For a more complete treatment of the holomorphic formalism, we refer to [18] (especially section 2 and appendix A); a general review of the superstrata solutions can be found in [19] (especially section 4.3).

The superstrata geometries are supersymmetric solutions of six-dimensional minimal supergravity coupled to two tensor multiplets. The bosonic fields are a metric, three three-form field strengths satisfying certain self-duality relations, and two scalars [27, 28]. The six-dimensional metric, using coordinates $u, v, r, \theta, \varphi_1, \varphi_2$, is given by:

$$ds_6^2 = -\frac{2}{\sqrt{\mathcal{P}}}(dv + \beta)\Big[du + \omega + \frac{\mathcal{F}}{2}(dv + \beta)\Big] + \sqrt{\mathcal{P}}\, ds_4^2, \tag{1}$$

where:

$$ds_4^2 = \Sigma\left(\frac{dr^2}{r^2+a^2}+d\theta^2\right)+(r^2+a^2)\sin^2\theta\,d\varphi_1^2+r^2\cos^2\theta\,d\varphi_2^2, \qquad \Sigma \equiv (r^2+a^2\cos^2\theta),$$

$$\beta \equiv \frac{R_y\,a^2}{\sqrt{2}\,\Sigma}(\sin^2\theta\,d\varphi_1-\cos^2\theta\,d\varphi_2), \qquad \mathcal{P} = Z_1 Z_2 - Z_4^2. \tag{2}$$

The six-dimensional coordinates $u, v$ are related to the time coordinate $t$ and a compact $y$ coordinate with $y \sim y + 2\pi R_y$ as:

$$u = \frac{1}{\sqrt{2}}(t-y), \qquad v = \frac{1}{\sqrt{2}}(t+y). \tag{3}$$

Besides $R_y$, the solution depends also on the constants $a$ (related to the five-dimensional angular momenta), and $Q_1, Q_5$ (the D1 and D5 charges of the solution).

The six-dimensional solution is determined by specifying three scalar functions $Z_1, Z_2, Z_4$, three two-forms $\Theta_1, \Theta_2, \Theta_4$ (which appear in the six-dimensional three-forms), as well as the metric one-form $\omega$ and the metric scalar function $\mathcal{F}$. The explicit expressions for $Z_I, \Theta_I$ can be found in eq. (6.9) in [18].

It is most convenient to use the following complex coordinates:

$$\xi \equiv \frac{r}{\sqrt{r^2+a^2}}e^{i\frac{\sqrt{2}v}{R_y}}, \quad \chi \equiv \frac{a}{\sqrt{r^2+a^2}}\sin\theta\,e^{i\varphi_1}, \quad \eta \equiv \frac{a}{\sqrt{r^2+a^2}}\cos\theta\,e^{i\left(\frac{\sqrt{2}v}{R_y}-\varphi_2\right)}, \quad (4)$$

which satisfy $|\xi|^2 + |\chi|^2 + |\eta|^2 = 1$.

A superstrata geometry is in principle completely determined by two arbitrary holomorphic functions of these three complex variables:[3]

$$G_1(\xi,\chi,\eta) \equiv \sum_{k,m,n} b_{k,m,n}\,\xi^n\,\chi^{k-m}\,\eta^m, \qquad G_2(\xi,\chi,\eta) \equiv \sum_{k,m,n} c_{k,m,n}\,\xi^n\,\chi^{k-m}\,\eta^m. \tag{5}$$

Note that $G_1(\xi,\chi,\eta)$ carries the so-called "original" ($q=0$) superstrata mode information and $G_2(\xi,\chi,\eta)$ carries the so-called "supercharged" ($q=1$) superstrata modes; see also section 4.1.

The metric warp factor is given by:

$$\mathcal{P} = Z_1 Z_2 - Z_4^2 = \frac{1}{\Sigma^2}\left(Q_1 Q_5 - \frac{R_y^2}{2}|G_1|^2\right). \tag{6}$$

The scalar function $\mathcal{F}$ and the one-form $\omega$ depend on the particular $G_1, G_2$ and can be extremely complicated expressions. In [18], the explicit expressions for $\mathcal{F}, \omega$ were found for certain families of superstrata; the general solution for arbitrary multimode $G_1, G_2$ is not known. For a general single mode (only one $b_{k,m,n}$ or $c_{k,m,n}$ non-zero) geometry, the solution can be found in e.g. section 4.3 of [19].

The superstrata geometries are three-charge geometries, carrying a D1-brane charge $Q_1$, D5-brane charge $Q_5$, and momentum ($P$) charge $Q_P$. This momentum charge, in the most general superstrata geometry, is most easily expressed in terms of the modes $b_{k,m,n}, c_{k,m,n}$:

$$Q_P = \sum_{k,m,n}\frac{m+n}{2k}(C_{k,m,n})^2\left(|b_{k,m,n}|^2 + \frac{k^2}{mn(k-m)(k+n)}|c_{k,m,n}|^2\right), \tag{7}$$

---

[3]The range of the integers is: $k \geq 1$, $0 \leq m \leq k$, and $n \geq 1$ for $G_1$ and $k \geq 1$, $1 \leq m \leq k-1$, and $n \geq 1$ for $G_2$ [29].

where we have defined the combinatorial factor:

$$C_{k,m,n} \equiv \left[ \binom{k}{m} \binom{k+n-1}{n} \right]^{-1/2}. \tag{8}$$

Finally, regularity forces the holomorphic functions to be constrained by the other parameters of the solution through [30]:

$$2 \left( \frac{Q_1 Q_5}{R_y^2} - a^2 \right) = \sum_{k,m,n} (C_{k,m,n})^2 \left( |b_{k,m,n}|^2 + \frac{k^2}{mn(k-m)(k+n)} |c_{k,m,n}|^2 \right). \tag{9}$$

## 2.2 $(1,0,n)$ superstrata

First, we turn our attention to perhaps the simplest family of superstrata: the so-called $(1,0,n)$ solutions. This family of solutions has the advantage of being explicitly known for any (multimode) solution, and in addition it can be reduced to three dimensions (which we will use in section 3 to calculate the symplectic form in supergravity explicitly). We will first review the $(1,0,n)$ solutions below, before deriving their symplectic form.

### 2.2.1 The solutions

The $(1,0,n)$ family of superstrata has $G_2 = 0$ and $G_1 = \chi F(\xi)$ in (5), with $F$ an arbitrary holomorphic function:

$$F(\xi) = \sum_{n=1}^{\infty} b_n \xi^n, \tag{10}$$

which satisfies $F(0) = 0$ and its complex conjugate is $\bar{F} \equiv \bar{F}(\bar{\xi})$. The metric functions are then given by the simple expressions:

$$\mathcal{F} = \frac{1}{a^2}(|F|^2 - |F_\infty|^2), \tag{11}$$

$$\omega = \left( 1 - \frac{1}{2a^2}(|F_\infty|^2 - c) \right) \omega_0 + \frac{R_y}{\sqrt{2}\Sigma}(|F_\infty|^2 - |F|^2) \sin^2\theta \, d\varphi_1,$$

$$\omega_0 = \frac{a^2 R_y}{\sqrt{2}\Sigma}(\sin^2\theta \, d\varphi_1 + \cos^2\theta \, d\varphi_2),$$

where we have defined:

$$\xi_\infty := \lim_{r \to \infty} \xi = e^{i\frac{\sqrt{2}v}{R_y}}, \qquad F_\infty := F(\xi_\infty). \tag{12}$$

The function $F$ and constant $c$ must satisfy the constraint (9), which for the $(1,0,n)$ family reads:

$$c = 2 \left( \frac{Q_1 Q_5}{R_y^2} - a^2 \right) = \frac{1}{\sqrt{2}\pi R_y} \int_0^{\sqrt{2}\pi R_y} dv' |F_\infty|^2 = \sum_{n=1}^{\infty} |b_n|^2. \tag{13}$$

For more details, see [18] (sections 2.5 and 3.1) and [26] (appendix D, especially D.5). These solutions are completely regular for any choice of $F$ [18]. The momentum charge (7) can be expressed as a sum over modes or as a particular integral involving $F$:

$$Q_P = \frac{1}{4\sqrt{2}\pi R_y} \int_0^{\sqrt{2}\pi R_y} dv(\xi_\infty F'_\infty \bar{F}_\infty + \bar{\xi}_\infty F_\infty \bar{F}'_\infty) = \frac{1}{2} \sum_{n=1}^{\infty} n|b_n|^2. \tag{14}$$

### 2.2.2  The symplectic form

The D1-D5-P superstrata are supersymmetric, so the Hamiltonian is quite simply (in units where $G_5 = \pi/4$, see section 3.2):

$$H = Q_1 + Q_5 + Q_P, \tag{15}$$

where $Q_P$ is given by (14). Noting that the derivative in the integral in (14) is with respect to $\xi$, we can rewrite the Hamiltonian in a simpler form involving only $v$ and using partial integration together with the periodicity condition $F_\infty(v = 0) = F_\infty(v = \sqrt{2}\pi R_y)$, giving:

$$H = Q_1 + Q_5 + \frac{1}{4\pi i} \int dv\, \bar{F}_\infty \partial_v F_\infty. \tag{16}$$

From this expression, it is clear that $F_\infty, \bar{F}_\infty$ will be the coordinates on the phase space.[4] Now, using the relation (3) between $v$ and the time coordinate $t$, the time-dependence of the geometry must be given by:

$$\frac{d}{dt} F_\infty(v) = \frac{i}{R_y} F'_\infty(v) \xi_\infty = \frac{1}{\sqrt{2}} \partial_v F_\infty. \tag{17}$$

However, we can also use the fundamental relation (115) involving the symplectic form:

$$\frac{d}{dt} F_\infty(v) = \{F_\infty, H\}_{\mathrm{PB}} = \omega^{F\bar{F}} \frac{1}{4\pi i} \partial_v F_\infty. \tag{18}$$

From (17) and (18) it follows that $\omega^{F\bar{F}} = 2\sqrt{2}\pi i$ and thus the symplectic form is:

$$\Omega = \frac{i\sqrt{2}}{4\pi} \int dv\, \delta F_\infty \wedge \delta \bar{F}_\infty. \tag{19}$$

Note that the fundamental Poisson bracket is:

$$\{F_\infty(v), \bar{F}_\infty(v')\}_{\mathrm{PB}} = i 2\sqrt{2}\pi \delta(v - v'). \tag{20}$$

We can also express the symplectic form and Poisson bracket in terms of the oscillators $b_n$. Noting that:

$$b_n = \frac{1}{\sqrt{2}\pi R_y} \int dv\, F_\infty(v) e^{-ni \frac{\sqrt{2}}{R_y} v}, \tag{21}$$

we integrate the Poisson bracket (20) to get:

$$\{b_n, \bar{b}_m\}_{\mathrm{PB}} = i \delta_{mn} \frac{2}{R_y}. \tag{22}$$

So, we recognize that actually the rescaled operators

$$\hat{b}_m \equiv \sqrt{\frac{R_y}{2}} b_m, \tag{23}$$

are those that satisfy the canonical commutators:

$$\{\hat{b}_n, \bar{\hat{b}}_m\}_{\mathrm{PB}} = i \delta_{mn}. \tag{24}$$

---

[4]In principle, it is possible that we would also need to include derivatives (with respect to $v$) of $F_\infty, \bar{F}_\infty$ as coordinates in the symplectic form, just as the D1-D5 supertube symplectic form (124) contains both $\vec{F}(s)$ and $\vec{F}'(s)$. One could redo the analysis allowing for this possibility, but *a posteriori* it is clear that only considering $F_\infty, \bar{F}_\infty$ as phase space coordinates is sufficient.

Note that the time-dependence of $b_n$ is simply given by:

$$\frac{d}{dt}b_n = i\frac{n}{R_y}b_n, \qquad b_n(t) = b_n(t=0)e^{i\frac{n}{R_y}t},$$
(25)

which (as should be expected) is simply the time dependence of the $b_n\xi_\infty^n$ term in $F(\xi_\infty)$ in (10).

We see that using the Rychkov consistency condition, we are easily able to find the supergravity symplectic form of the $(1,0,n)$ superstrata; we will further confirm this by an explicit calculation in section 3. Note that the consistency condition for the D1-D5 system was only enough to find the symplectic form up to an overall constant (see appendix B) which then required an explicit calculation to find; by contrast, for the superstrata, we are able to obtain the *entire* symplectic form directly from the consistency condition.

### 2.3 $(k,m,n)$ general superstrata

The solution for the most generic superstrata with multiple original and supercharged modes turned on is not explicitly known. However, the above analysis of the $(1,0,n)$ subfamily of superstrata has taught us that the only ingredients necessary to find the symplectic form in supergravity are the Hamiltonian in terms of the modes and the expected time-dependence of the modes. Thus, we will easily be able to generalize the above analysis to find the supergravity symplectic form for the (at this moment, strictly speaking, *hypothetical*) general multimode superstrata geometry.

The Hamiltonian is still given by the sum of charges (15), but now the momentum charge $Q_P$ is given by the more complicated expression (7). For the general superstrata, it is more convenient to work directly with the oscillators directly when applying the Rychkov consistency condition, since an expression for $Q_P$ in terms of the holomorphic functions (5) would be too unwieldy.

Focusing on a single $b_{k,m,n}$, the required time-dependence can be read off simply from (4) and (5), which gives (generalizing (25)):

$$\frac{d}{dt}b_{k,m,n} = i\frac{n+m}{R_y}b_{k,m,n},$$
(26)

whereas from the Hamiltonian and (7), it follows that:

$$\frac{d}{dt}\left(\log b_{k,m,n}\right) = \frac{1}{2}(C_{k,m,n})^2\frac{m+n}{k}\{b_{k,m,n}, \bar{b}_{k,m,n}\}_{\text{PB}}.$$
(27)

From this, we can immediately and easily read off the Poisson bracket:

$$\{b_{k,m,n}, \bar{b}_{k',m',n'}\}_{\text{PB}} = i\,\delta_{kk'}\delta_{mm'}\delta_{nn'}\frac{k}{(C_{k,m,n})^2}\frac{2}{R_y}.$$
(28)

Again, we can rescale the oscillators to:

$$\hat{b}_{k,m,n} \equiv \sqrt{\frac{R_y}{2k}}C_{k,m,n}b_{k,m,n},$$
(29)

which satisfy the canonical bracket:

$$\{\hat{b}_{k,m,n}, \hat{\bar{b}}_{k',m',n'}\}_{\text{PB}} = i\delta_{kk'}\delta_{mm'}\delta_{nn'}.$$
(30)

These expressions (29) and (30) generalize the $(1,0,n)$ results (23) and (24) above.

The analysis of the supercharged modes $c_{k,m,n}$ proceeds in a precisely analogous way, and leads to the rescaled oscillators:

$$\hat{c}_{k,m,n} \equiv \sqrt{\frac{R_y}{2}} C_{k,m,n} \sqrt{\frac{k}{mn(k-m)(k+n)}} c_{k,m,n}, \tag{31}$$

which satisfy the canonical bracket:

$$\{\hat{c}_{k,m,n}, \hat{\bar{c}}_{k',m',n'}\}_{\text{PB}} = i \delta_{kk'} \delta_{mm'} \delta_{nn'}. \tag{32}$$

Finally, for completeness, we state the resulting total symplectic form for the most general superstrata:

$$\Omega = i \left( \sum_{k,m,n} \delta \hat{b}_{k,m,n} \wedge \delta \hat{\bar{b}}_{k,m,n} + \sum_{k,m,n} \delta \hat{c}_{k,m,n} \wedge \delta \hat{\bar{c}}_{k,m,n} \right), \tag{33}$$

where we used the mode expansions (5) and the rescaled operators (29) and (31).

# 3 Explicit Supergravity Computation for $(1, 0, n)$

Recently, it was found that the generic $(1, 0, n)$ superstrata (as well as the more general $(1, m, n)$ superstrata) can be dimensionally reduced from six to three dimensions [26]. As we show here, this dimensional reduction makes it possible to explicitly calculate the symplectic form for these superstrata very easily using the standard methods in supergravity. This allows us to explicitly confirm the symplectic form (19) as found above using Rychkov's consistency condition.

## 3.1 The $(1, 0, n)$ superstrata in 3D

We have already introduced the $(1, 0, n)$ superstrata in a six-dimensional form in sections 2.1 and 2.2.1. Here, we will briefly review the Lagrangian and solution for the general $(1, 0, n)$ superstrata when we reduce the solution to three dimensions (as discussed in sections 3.4 & 4.3 of [26]).

The bosonic sector of the relevant three-dimensional supergravity theory contains the metric, 6 scalars $\xi_1, \xi_2, \xi_3, \xi_4, \chi_1, \chi_2$, and two $U(1)$ gauge fields $A^{\varphi_1}, A^{\varphi_2}$. The Lagrangian is [26]:

$$\mathcal{L}_{3D,U(1)^2} = R - \frac{1}{2}(\partial_\mu \xi_1)^2 - \frac{1}{2}(\partial_\mu \xi_2)^2 - \frac{1}{2}(\partial_\mu \xi_3)^2 - \frac{1}{2}\sinh^2 \xi_3 (\mathcal{D}_\mu \xi_4)^2 \tag{34}$$

$$- \frac{1}{4} e^{-2\xi_1} F^{\varphi_1}_{\mu\nu} F^{\varphi_1,\mu\nu} - \frac{1}{4} e^{-2\xi_2} F^{\varphi_2}_{\mu\nu} F^{\varphi_2,\mu\nu} - \frac{1}{2} e^{\xi_2} \left( \cosh \xi_3 \left[ (\mathcal{D}_\mu \chi_1)^2 + (\mathcal{D}_\mu \chi_2)^2 \right] \right.$$

$$- \sinh \xi_3 \left[ \sin \xi_4 \left( (\mathcal{D}_\mu \chi_1)^2 - (\mathcal{D}_\mu \chi_2)^2 \right) + 2 \cos \xi_4 \mathcal{D}_\mu \chi_1 \mathcal{D}^\mu \chi_2 \right] \right)$$

$$+ e^{-1} \epsilon^{\mu\nu\rho} \left( 2\alpha A^{\varphi_1}_\mu F^{\varphi_2}_{\nu\rho} + \frac{1}{4} \varepsilon F^{\varphi_2}_{\mu\nu} (\chi_2 \mathcal{D}_\rho \chi_1 - \chi_1 \mathcal{D}_\rho \chi_2) \right) - V,$$

where the scalar potential is given by:

$$V = -2g_0^2 e^{\xi_1} \left( 2 e^{\xi_2} \cosh \xi_3 - e^{\xi_1} \sinh^2 \xi_3 \right) + \frac{g_0^2}{2} e^{2\xi_1 + \xi_2} \left[ e^{\xi_2} \left( \frac{1}{2} \varepsilon \chi_1^2 + \frac{1}{2} \varepsilon \chi_2^2 + 4 g_0^{-1} \alpha \right)^2 \right.$$

$$+ \cosh \xi_3 \left( \chi_1^2 + \chi_2^2 \right) + \sinh \xi_3 \left( (\chi_1^2 - \chi_2^2) \sin \xi_4 + 2 \chi_1 \chi_2 \cos \xi_4 \right) \right], \tag{35}$$

and the gauge-covariant derivatives are:

$$\mathcal{D}_\mu \chi_1 = \partial_\mu \chi_1 + g_0 \chi_2 A^{\varphi_1}_\mu, \qquad\qquad \mathcal{D}_\mu \chi_2 = \partial_\mu \chi_2 - g_0 \chi_1 A^{\varphi_1}_\mu, \tag{36}$$

$$\mathcal{D}_\mu \xi_4 = \partial_\mu \xi_4 + 2 g_0 A^{\varphi_1}_\mu. \tag{37}$$

A series of rescalings can take $\alpha, g_0$ to any value we wish [26]; it is most convenient to choose:

$$\alpha = -\frac{1}{2}\varepsilon g_0, \qquad g_0 = (Q_1 Q_5)^{-1/4}, \tag{38}$$

so that $g_0^{-1}$ is the radius of the $S^3$ in the six-dimensional uplift appropriate for a D1-D5-P superstrata. The Lagrangian (34) also depends on the sign $\varepsilon = \pm 1$, which is related to the supersymmetry of the solution [26, 31].

The general (multimode) $(1, 0, n)$ superstrata solution in this three-dimensional system can be given in the coordinates $(u, v, r)$, where $u, v$ are related to $t, y$ as in (3). In particular, recall that $y$ is periodic with radius $R_y$. It is often convenient to package the coordinates $v, r$ into the complex coordinate $\xi$ given in (4). We are required to take the orientation [26]:

$$e^{-1}\epsilon_{uvr} = -\varepsilon. \tag{39}$$

This is the only place that the sign $\varepsilon$ shows up in the solution.

We repeat that the $(1, 0, n)$ solution is then completely determined by an arbitrary holomorphic function (10) of this coordinate [18]:

$$F \equiv F(\xi) = \sum_{n=1}^{\infty} b_n \xi^n, \tag{40}$$

that satisfies $F(0) = 0$.

Explicitly, the $(1, 0, n)$ solution reduced to three dimensions is given by the solution of the Lagrangian (34) with $\xi_1 = \xi_3 = \xi_4 = 0$ and [26]:

$$\chi_{1,2} = 2S_{1,2}, \tag{41}$$

with

$$S_1 = -\frac{iaR_y g_0^2}{2\sqrt{2(a^2+r^2)}}\left(F - \bar{F}\right) \qquad \text{and} \qquad S_2 = -\frac{aR_y g_0^2}{2\sqrt{2(a^2+r^2)}}\left(F + \bar{F}\right). \tag{42}$$

The three dimensional metric, $ds_3^2$, takes the form:[5]

$$ds_3^2 = \frac{R_y^2 g_0^2}{2}\left[\Xi^2 ds_2^2 - a^4 g_0^4\left(du + dv + \frac{\sqrt{2}}{a^2 R_y g_0^4}\mathscr{A}\right)^2\right], \tag{43}$$

where:

$$ds_2^2 = \frac{|d\xi|^2}{(1-|\xi|^2)^2}, \qquad \Xi^2 = \frac{2}{R_y^2 g_0^4}(1 - S_A S_A), \qquad \mathscr{A} = \frac{i}{2}\left(\frac{\xi d\bar{\xi} - \bar{\xi}d\xi}{1-|\xi|^2}\right), \tag{44}$$

and:

$$\Xi^2 = \frac{2}{R_y^2 g_0^4}(1 - S_1^2 - S_2^2). \tag{45}$$

The remaining scalar is:

$$e^{-\xi_2} = \frac{1}{2}R_y^2 g_0^4 \Xi^2. \tag{46}$$

---

[5]Note that we would need to perform a large gauge transformation on (43) to put it in a form which is asymptotically $AdS_3$ (see [26], appendix D.3), which is the gauge in which it is given in (11). This distinction using the gauge transformation will not be necessary or important for our calculations.

The vector fields are:

$$A_\mu^{\varphi_1} \, dx^\mu = -\frac{a^2 R_y g_0^3}{\sqrt{2}}(du + dv), \tag{47}$$

$$A_\mu^{\varphi_2} \, dx^\mu = \frac{\sqrt{2}}{R_y g_0 \Xi^2}\left[ a^2(du + dv) + \frac{2}{a^2 R_y^2 g_0^4}\left((a^2 + r^2)(S_1^2 + S_2^2) - a^2\right) dv \right]. \tag{48}$$

Note that all fields except the scalars $\chi_{1,2}$ only depend on $F, \bar{F}$ through the combination $\Xi^2$.

The parameters of this solution are the same as those of the six-dimensional solution in section 2.2.1: the D1 and D5 charges $Q_1, Q_5$ (through (38)), the angular momentum parameter $a$, and the radius $R_y$ of the $y$-circle. Recall that the parameters must satisfy the constraint (13), and that the momentum P charge is given by $Q_P$ in (14).

## 3.2 The symplectic form from 3D supergravity

To find the symplectic form in supergravity, we first calculate the symplectic current using the standard formalism of [24] (see also [23]). The general semi-classical symplectic form for a theory with Lagrangian $L$ is:

$$J^\mu = \sum_A \delta\left(\frac{\partial L}{\partial(\partial_\mu \phi^A)}\right) \wedge \delta \phi^A, \tag{49}$$

where the sum is over all (fundamental) fields $\phi^A$ in the theory. Note that $L$ includes the prefactor of $\sqrt{-g}$, so the action (in three dimensions) is simply:

$$S = \int d^3x \, L. \tag{50}$$

The symplectic form is then given by:

$$\Omega = \int_\Sigma d\Sigma_\mu J^\mu, \tag{51}$$

where we integrate the symplectic current over a Cauchy surface $\Sigma$.

The symplectic current is an object that lives on the *solution phase space*, which means the variations considered in (49) are *on-shell*. In other words, the fields $\phi^A$ as well as $\phi^A + \delta \phi^A$ always solve the equations of motion. We can then further restrict the phase space to the family of solutions we are interested in — in this case, the $(1, 0, n)$ superstrata.

Since the $(1, 0, n)$ superstrata solutions can be reduced to three dimensions, the symplectic form as calculated with the three-dimensional effective action (34) gives the same result as the calculation in the full ten-dimensional supergravity would. Specifically, the three-dimensional theory is obtained from six dimensions by reducing on an $S^3$ with radius $g_0^{-1}$ [26], which in turn is obtained from ten dimensions by reducing over a $T^4$ with volume $V_4$ [27, 28], so the various Newton constants are related by:

$$G_3 = \frac{G_{10}}{V_4 \, \text{vol}(S^3)} = \frac{G_{10}}{V_4 \, 2\pi^2 g_0^{-3}}. \tag{52}$$

We are working in units where in a five-dimensional frame, obtained from ten dimensions by reducing over the same $T^4$ and then also reducing over the $S^1$ parametrized by $y$, the Newton constant is given by:

$$G_5 = \frac{G_{10}}{V_4(2\pi R_y)} = \frac{\pi}{4}. \tag{53}$$

This choice of units follows since we want the five-dimensional mass to be given by $M_{5D} = Q_1 + Q_5 + Q_P$ [32]. Together, (52) and (53) imply that:

$$G_3 = \frac{R_y g_0^3}{4}. \tag{54}$$

Note that the action of our three-dimensional theory is given by:

$$S = \frac{1}{16\pi G_3} \int d^3x \sqrt{-g}\, \mathcal{L}_{3D,U(1)^2}, \tag{55}$$

with the Lagrangian given in (34).

### 3.3 Calculating the $(1,0,n)$ symplectic form

As we discussed above, a $(1,0,n)$ superstrata geometry is entirely determined by the holomorphic function $F(\xi)$, as detailed above in section 3.1. A perturbation in the solution space, $\phi^A \to \phi^A + \delta\phi^A$, is generated by perturbing this function, $F \to F + \delta F$. This simplifies the calculation, since the only fields of the solution that change when perturbing $F$ are the metric $g_{\mu\nu}$, the gauge field $A^{\varphi_2}$, and the scalars $\xi_2, \chi_1, \chi_2$; thus, these are the only fields we need to consider in the sum over fields in (49). We can now explicitly calculate each of their contributions to (49), using the Lagrangian (34) and the solution of section 3.1.

**Metric $g_{\mu\nu}$**   The metric symplectic current is the Crnkovic-Witten current [23–25]:

$$16\pi G_3 J_g^\mu = -\delta\Gamma_{\nu\rho}^\mu \wedge \delta(\sqrt{-g}\, g^{\nu\rho}) + \delta\Gamma_{\nu\rho}^\rho \wedge \delta(\sqrt{-g}\, g^{\mu\nu}). \tag{56}$$

Note that the metric (43) only depends on $F$ (and $\bar{F}$) through the combination $\Xi$. After explicit evaluation, we simply find:

$$J_g^\mu = 0. \tag{57}$$

**Scalar $\xi_2$**   Note that $\xi_2$ in (46) also only depends on $F, \bar{F}$ through $\Xi$. We find:

$$16\pi G_3 J_{\xi_2}^\mu = \delta\left(\sqrt{-g}\,[-\partial^\mu \xi_2]\right) \wedge \delta\xi_2 \tag{58}$$

$$= \left(-\frac{\delta\Xi^2 \wedge \delta\partial_\nu\Xi^2 \left(a^4 g_0^4 R_y^2 + 2r^2\right)}{2g_0 r\left(a^2 + r^2\right)\Xi^4}, \frac{a^4 g_0^3 R_y^2 \delta\Xi^2 \wedge \delta\partial_\nu\Xi^2}{2r\left(a^2 + r^2\right)\Xi^4}, \frac{g_0^3 r\left(a^2 + r^2\right)\delta\Xi^2 \wedge \delta\partial_r\Xi^2}{\Xi^4}\right).$$

**Gauge field $A_\mu^{\varphi_2}$**   Note that $A^{\varphi_2}$ in (48) only depends on $\Xi$, but its action depends explicitly on $F$ through the scalars $\chi_{1,2}$. It is convenient to split the contribution from $A^{\varphi_2}$ into two parts:

$$J_{A^{\varphi_2}}^\mu = J_{A^{\varphi_2},(1)}^\mu + J_{A^{\varphi_2},(2)}^\mu, \tag{59}$$

$$16\pi G_3 J_{A^{\varphi_2},(1)}^\mu := \delta\left(\sqrt{-g}\left[-e^{-2\xi_2} F^{\varphi_2,\mu\nu} + 4\alpha e^{-1}\epsilon^{\mu\nu\rho} A_\rho^{\varphi_1}\right]\right) \wedge \delta A_\mu^{\varphi_2} = -J_{\xi_2}^\mu, \tag{60}$$

$$16\pi G_3 J_{A^{\varphi_2},(2)}^\mu := \delta\left(\sqrt{-g}\left[\frac{1}{2}e^{-1}\epsilon^{\mu\nu\rho}(\chi_2 \mathcal{D}_\rho \chi_1 - \chi_1 \mathcal{D}_\rho \chi_2)\right]\right) \wedge \delta A_\mu^{\varphi_2}. \tag{61}$$

The first contribution $J_{A^{\varphi_2},(1)}^\mu$ to the symplectic current cancels the contribution of $\xi_2$, so we are only left with the contribution of $J_{A^{\varphi_2},(2)}^\mu$. We do not give this expression here, since it is rather lengthy and unilluminating.

**Scalars $\chi_{1,2}$**   The symplectic form contributions are:

$$16\pi G_3 J^\mu_{\chi_1} = \delta\left(\sqrt{-g}\left[-e^{\xi_2}D^\mu\chi_1 + \frac{1}{4}\chi_2 e^{-1}\epsilon^{\nu\rho\mu}F^{\varphi_2}_{\nu\rho}\right]\right)\wedge\delta\chi_1 \tag{62}$$

$$16\pi G_3 J^\mu_{\chi_2} = \delta\left(\sqrt{-g}\left[-e^{\xi_2}D^\mu\chi_2 - \frac{1}{4}\chi_1 e^{-1}\epsilon^{\nu\rho\mu}F^{\varphi_2}_{\nu\rho}\right]\right)\wedge\delta\chi_2. \tag{63}$$

We again choose not to write these expressions explicitly as they are unilluminating.

**Total symplectic current**   Putting the pieces together from above, the full expression for the symplectic current will be given by:

$$J^\mu = J^\mu_{A^{\varphi_2},(2)} + J^\mu_{\chi_1} + J^\mu_{\chi_2}. \tag{64}$$

Since it is a symplectic vector (density), it satisfies:[6]

$$\partial_\mu J^\mu = 0, \tag{65}$$

and so we can find a symplectic potential $K^{\mu\nu}$, such that:

$$J^\mu = \partial_\nu K^{\mu\nu}. \tag{66}$$

It is easiest to express this potential in $(u, \xi, \bar\xi)$ coordinates; we find:

$$16\pi G_3 K^{\xi\bar\xi} = \frac{2i\sqrt{2}a^2 g_0^7 R_y^3(|\xi|^2 - 1)}{(2 - g_0^4 R_y^2(1 - |\xi|^2)|F(\xi)|^2)^2}\delta F(\xi)\wedge\delta\bar F(\bar\xi), \tag{67}$$

$$16\pi G_3 K^{\xi u} = -g_0^3 R_y^2 \frac{a^2 g_0^4(1 - |\xi|^2)R_y^2 + 2|\xi|^2}{\bar\xi(2 - g_0^4 R_y^2(1 - |\xi|^2)|F(\xi)|^2)^2}\delta F(\xi)\wedge\delta\bar F(\bar\xi), \tag{68}$$

$$K^{\bar\xi u} = \left(K^{\xi u}\right)^*. \tag{69}$$

Note that $J^\mu$ is completely regular, so we do not require a regularizing gauge transformation to accompany the bare, "naive" variation of the solution, as opposed to the situation in e.g. [9,24]. In particular, notice that $K^{\mu\nu}$ vanishes at the "origin" $r = 0$, since $F(0) = 0$ and $\delta F(0) = 0$.

Finally, to get the symplectic form $\Omega$, we integrate the current $J^\mu$ over the Cauchy surface $\Sigma$ defined by $u = cte$:

$$\Omega = \int_\Sigma d\Sigma_\mu J^\mu = \int d\xi d\bar\xi J^u. \tag{70}$$

Using (66), we can convert this into a surface integral over the boundary $\partial\Sigma$ at $r \to \infty$:

$$\Omega = \frac{1}{2}\int_{\partial\Sigma} d\Sigma_{\mu\nu}K^{\mu\nu} \tag{71}$$

$$= -\int_{|\xi|=1} d\xi K^{u\bar\xi} + \int_{|\xi|=1} d\bar\xi K^{u\xi} \tag{72}$$

$$= \frac{1}{16\pi G_3}(i\sqrt{2}g_0^3 R_y)\int dv\, \delta F_\infty(v)\wedge\delta\bar F_\infty(v), \tag{73}$$

where we used that e.g. $d\xi = i\sqrt{2}/R_y \xi_\infty dv$. Finally, we conclude that:

$$\Omega = \frac{i\sqrt{2}}{4\pi}\int dv\, \delta F_\infty\wedge\delta\bar F_\infty, \tag{74}$$

which is precisely the symplectic form we found above in (19).

---

[6]Note that including the factor of $\sqrt{-g}$ in the calculations implies $J^\mu$ is a vector density instead of a vector.

# 4 Counting Superstrata

The superstrata geometry is completely determined by two holomorphic functions of three variables, (5), which can be taken as the coordinates of the phase space as we showed in section 2. Using Rychkov's consistency condition (section 2) and also by explicit computation in supergravity (section 3), we derived the phase space symplectic form for superstrata and showed that the Poisson bracket between the mode coefficients are simply given by (24), (30), and (32). In this section, we pass to quantum mechanics by replacing Poisson bracket by commutators, enabling us to count the number of superstrata states available for given D1, D5, and P charges. In our units (53), the charges $Q_1, Q_5, Q_P$ are related to the quantized numbers of branes $N_1, N_5, N_P$ by

$$\frac{Q_1 Q_5}{R_y} = N_1 N_5 =: N, \qquad R_y Q_P = N_P. \tag{75}$$

We are interested in counting these superstrata in the regime $N, N_P \gg 1$.

Counting of superstrata has already been done in [22] from the CFT side, and for states that correspond to superstrata more general than are discussed in the current paper. In that sense, the counting presented in this section is not new. Here, for the generic $(k, m, n)$ superstrata, we will reproduce (see (103)) the entropy growth $S \sim N^{1/4} N_P^{1/2}$ (for $N_P \gg N$) found in [22], using the symplectic form obtained in the previous sections from supergravity. The difference with the calculation in [22], besides being done here from the gravity side, is that we restrict to a simple subsector (the original superstrata based on $|00\rangle$) instead of the general superstrata counted in [22], and that we consider ensembles characterized only by $N, N_P$; we ignore the R-charge $J := J_0^3 = m$ which corresponds to left-moving angular momentum in six dimensions.

## 4.1 CFT dual

So far we have been discussing superstrata solutions in supergravity and their phase space. Here we very briefly describe their CFT dual, for it is useful in understanding the structure of phase and Hilbert spaces that we have in supergravity.

The AdS/CFT dual of our six- or three-dimensional gravity is a two-dimensional orbifold CFT with target space $(T^4)^N / S_N$, called the D1-D5 CFT.[7] The states in the Hilbert space of this theory can be thought of as made of strings or "strands". A strand of length $k$ represents $k$ copies of $T^4$ intertwined with each other by the orbifold action. Because we have $N$ copies of $T^4$, the total length of all the strands must be equal to $N$. Strands come in multiple flavors and the ones relevant here are denoted by $|++\rangle_k$ and $|00\rangle_k$, where $k$ ($1 \le k \le N$) is the length of the strand. These strands are generically 1/4-BPS (preserving 8 supercharges). Empty AdS$_3$ space corresponds to $[|++\rangle_1]^N$, while the D1-D5 geometries [5,6] counted by Rychkov [9] (see also appendix B) correspond to additionally considering 1/4-BPS strands with different flavors and lengths.

We can excite various modes on these strands. In particular, by acting on these strands with generators of the superconformal algebra, we can construct 1/8-BPS excitations denoted by $|k, m, n, q = 0\rangle$ and $|k, m, n, q = 1\rangle$,[8] which are in direct correspondence with the $(k, m, n)$ family of superstrata (original and supercharged, respectively). The 1/8-BPS strand $|k, m, n, q\rangle$ has length $k$ and left-moving momentum $m + n$. (It also has R-charge $J := J_0^3 = m$.)

Assume that we start with $N |++\rangle_1$ strands (representing empty AdS$_3$) and replace some of them with the excited strands $|k, m, n, q = 0\rangle$ with various $k, m, n$. This corresponds to exciting

---

[7]For more detail about the D1-D5 CFT, see e.g. [33, 34].

[8]More explicitly, $|k, m, n, q = 0\rangle = (J_{-1}^+)^m (L_{-1} - J_{-1}^3)^n |00\rangle_k$, $|00, k, m, n, q = 1\rangle = (J_{-1}^+)^{m-1} (L_{-1} - J_{-1}^3)^{n-1} (G_{-1/2}^{+,1} G_{-1/2}^{+,2} + (1/2h^{NS})(L_{-1} - J_{-1}^3) J_{-1}^+) |00\rangle_k$, where $L_n, J_n^i, G_n^{\alpha A}$ are generators of the superconformal symmetry $SU(1,1|2)_L$. See [19, 29, 30] for more detail.

the $(k, m, n)$ superstrata. If $N_{k,m,n}$ is the number of the $|k, m, n, q = 0\rangle$ strands and $N_0$ is the number of $|++\rangle_1$ strands, then we must demand that the total strand length remains fixed at $N$:

$$N_0 + \sum_{k,m,n} k N_{k,m,n} = N. \tag{76}$$

In supergravity, this condition appears as the regularity condition (9) in the geometry.

## 4.2 Counting $(1, 0, n)$

Let us come back to supergravity and count the $(1, 0, n)$ family of superstrata whose phase space structure was studied in sections 2.2 and 3. This is not a "natural" ensemble in the sense that this is not the most general family of supergravity solutions specified by the macroscopic charges $N$ and $N_P$; we are imposing by hand the condition that $k = m = 0$.[9] However, this is a simple, illustrative example that we can work out before discussing the more general case.

The Poisson bracket (24) for the mode coefficients $\hat{b}_n, \bar{\hat{b}}_m$ is replaced by the canonical bosonic quantum commutator

$$[\hat{b}_n, \hat{b}_m^\dagger] = \delta_{mn}. \tag{77}$$

From (14) and (75), the quantized momentum number $N_P$ is given by

$$N_P = \sum_{n=1}^{\infty} n N_n, \qquad N_n := \langle \hat{b}_n^\dagger \hat{b}_n \rangle, \tag{78}$$

where $N_n = 0, 1, 2, \ldots$ counts the excitation number of mode $n$. Therefore, counting the $(1, 0, n)$ family of superstrata amounts to counting possible partitions $\{N_n\}$ of the integer $N_P$. However, we must also take into account the additional constraint (76) on $N_n$ (equivalent to (13)), which implies:

$$\sum_{n=1}^{\infty} N_n \leq N. \tag{79}$$

In the dual CFT, this corresponds to the fact that the sum of the lengths of the excited strands $|1, 0, n, q = 0\rangle$ cannot exceed the total length $N$.

Our task is, for given $N$ and $N_P$, to count the partitions $\{N_n\}$ that satisfies (78) and (79). If $N_P$ is small compared to $N$ (note that also both $N, N_P \gg 1$), the constraint (79) is ineffective and the counting is that of a free chiral boson with energy $N_P$. How small should $N_P$ be for this to be valid? For a free boson, low ($n = \mathcal{O}(1)$) modes are most excited, with the excitation number $N_n \sim \sqrt{N_P}$. Thus the left-hand side of (79) is roughly $\sim \sqrt{N_P}$ and therefore this approximation is valid only for $\sqrt{N_P} \ll N$. Let us call this the "low-temperature" regime.[10] So, the entropy in the low-temperature regime is

$$S_{(1,0,n)} \approx 2\pi \sqrt{\frac{N_P}{6}}, \qquad N_P \ll N^2. \tag{80}$$

Let us confirm this by thermodynamic arguments. If we introduce $N_0 \geq 0$, equations (78)

---

[9]Also, we are restricting ourselves to states that are based on the strand of the special flavor, $|00\rangle$, among all possible flavors.

[10]Superstrata are supersymmetric and the physical temperature is zero. Here we are talking about the "temperature" conjugate to $N_P$ regarded as energy.

and (79) can be written as

$$\sum_{n=0}^{\infty} n N_n = N_P, \tag{81}$$

$$\sum_{n=0}^{\infty} N_n = N. \tag{82}$$

So, mode $n$ contributes $n$ and $1$ to $N_P$ and $N$, respectively. In CFT, $N_0$ represents the number of the ground-state strands, $|++\rangle_1$. Eq. (82) corresponds to the strand-length budget constraint (76) in CFT, with the identification $N_n = N_{1,0,n}$.

If we define fugacities by

$$p = e^{-\alpha}, \qquad q = e^{-\beta}, \tag{83}$$

we can write down a grand-canonical partition function

$$Z(p,q) = \sum_{N,N_P} c(N, N_P) p^N q^{N_P} = \frac{1}{(1-p)^{c_0}} \prod_{n=1}^{\infty} \frac{1}{1 - pq^n}, \tag{84}$$

from which we can read off the number of states $c(N, N_P)$. Here $c_0 = 1$ is the number of species of the $n = 0$ mode ($|++\rangle_1$ in CFT). If we also allow the $(1,0,0)$ superstratum, which is dual to $|1,0,0,q=0\rangle = |00\rangle_1$, we should set $c_0 = 2$. However, the value of $c_0$ does not matter to the final entropy. By using the formula $-\log(1-x) = \sum_{r=0}^{\infty} x^r/r$ and carrying out the summation over $n$, we find

$$\log Z = -c_0 \log(1-p) - \sum_{n=1}^{\infty} \log(1 - pq^n)$$
$$= -c_0 \log(1-p) + \sum_{r=1}^{\infty} \frac{p^r q^r}{r(1-q^r)}. \tag{85}$$

The low-temperature regime corresponds to $0 < \alpha \ll \beta \ll 1$. In this case, we can approximate the sum in (85) as

$$\log Z \approx -c_0 \log(1-p) + \sum_{r=1}^{\infty} \frac{1}{\beta r^2} \approx -c_0 \log \alpha + \frac{\pi^2}{6\beta}. \tag{86}$$

Then, we use the thermodynamical relations:

$$N = -\partial_\alpha \log Z = \frac{c_0}{\alpha}, \qquad \alpha = \frac{c_0}{N}, \tag{87}$$

$$N_p = -\partial_\beta \log Z = \frac{\pi^2}{6\beta^2}, \qquad \beta = \frac{\pi}{\sqrt{6N_p}}. \tag{88}$$

We can see that $0 < \alpha \ll \beta \ll 1$ indeed means that $N_P \ll N^2$. The entropy is

$$S_{(1,0,n)} = \log Z + \alpha N + \beta N_p \approx 2\pi \sqrt{\frac{N_p}{6}}, \tag{89}$$

which reproduces (80). The fact that $N$ depends only on $c_0, \alpha$ and not on $\beta$ means that the most of the system (which is of length $N$) is filled with the $n = 0$ modes which do not feel $\beta$. A negligibly small part (of length $\sim \sqrt{N_p} \ll N$) of the system is populated with $n > 0$ modes which are effectively free and responsible for the entropy (89).

In the opposite, "high-temperature" regime $N_p \gg N^2$, the picture is totally different. This corresponds to $0 < \beta \ll \alpha$. Actually, it turns out that $\beta \ll 1$ and $\alpha \gg 1$ (and therefore $p \ll 1$). Physically, this means that the cost $pq^n = e^{-\alpha - \beta n}$ to create an excitation in mode $n$ is almost the same for a wide range of $n$, $\Delta n \sim \alpha / \beta$. This allows modes with very large $n$ to be excited as easily as small $n$ modes, making it possible for large momentum to be carried by those large $n$ modes. Also, $p = e^{-\alpha} \ll 1$ means that only one quantum can be excited in each mode. So, in the high-temperature regime, $N_P$ is carried by a large number of modes with different values of $n$, each of which is excited only once. This in particular means that, in the partition function (85), the contribution from the $n = 0$ mode (the first term) is negligible compared to the contribution from other modes (the second term). Therefore, we can approximate the partition function as

$$\log Z \approx \sum_{r=1}^{\infty} \frac{p^r}{r(e^{\beta r} - 1)} \approx \frac{p}{e^{\beta} - 1} \approx \frac{p}{\beta}. \tag{90}$$

In the second "$\approx$", we only kept the $r = 1$ term because $p \ll 1$. From this, we find

$$N = p \, \partial_p \log Z = \frac{p}{\beta}, \qquad N_P = -\partial_{\beta} \log Z = \frac{p}{\beta^2}. \tag{91}$$

In other words,

$$p = \frac{N^2}{N_P}, \qquad \beta = \frac{N}{N_P}, \tag{92}$$

which indeed means that $p \ll 1$, $\beta \ll 1$, and $\beta \ll \alpha = -\log p$ if $N_P \gg N^2$. The entropy is computed to be

$$S_{(1,0,n)} = N \left( 2 + \log \frac{N_P}{N^2} \right), \qquad N_P \gg N^2. \tag{93}$$

We have numerically checked that this correctly reproduces the growth of $c(N, N_P)$ in this regime.

Finally, note that this high-temperature regime $N_P \gg N^2$ is outside of the regime of validity of the decoupling limit from which the AdS/CFT correspondence was derived [35]; the excitation is not confined within the near-brane region. Still, this counting is a well-defined problem with a clear physical interpretation, so is interesting in its own right.

## 4.3 Counting general $(k, m, n)$

Let us move on to counting $(k, m, n)$ strata. For simplicity, we focus on the original (and not the supercharged) superstrata. In [22], counting of $(k, m, n)$ superstrata, both original and supercharged, was done from the CFT side, not just for ones based on the $|00\rangle$ strand but also other flavors such as $|\pm\pm\rangle$.

Now, we have the occupation numbers $N_{k,m,n} := \langle \hat{b}^{\dagger}_{k,m,n} \hat{b}_{k,m,n} \rangle \geq 0$ that satisfy the constraint that comes from (7).

$$\sum_{k,m,n} (m + n) N_{k,m,n} = N_P. \tag{94}$$

In addition, (76) or equivalently (9) means that

$$\sum_{k,m,n} k N_{k,m,n} = N, \tag{95}$$

where, just like in the $(1, 0, n)$ case, we introduced $N_{k,0,0} \geq 0$, which does not carry $N_P$, to "fill" the Hilbert space of length $N$. The range of $k, m, n$ is: $k \geq 1$, $0 \leq m \leq k$, and $n \geq 1$.[11] The partition function is

$$Z(p, q) = \prod_{k=1}^{\infty} \prod_{m=0}^{k} \prod_{n=1}^{\infty} \frac{1}{(1 - p^k q^{m+n})}. \tag{96}$$

By similar manipulations as in (85), we can rewrite this as

$$\log Z = -\sum_{r=1}^{\infty} \frac{1}{r(1 - q^r)^2} \left( \frac{p^r q^r}{1 - p^r} - \frac{p^r q^{3r}}{1 - p^r q^r} \right). \tag{97}$$

Let us discuss the entropy of this system in the low- and high-temperature regimes, just as in the $(1, 0, n)$ case.

First, in the low-temperature regime defined by $0 < \alpha \sim \beta \ll 1$, we can approximate (97) by

$$\log Z \approx \sum_{r=1}^{\infty} \frac{1}{r(\beta r)^2} \left( \frac{1}{\alpha r} - \frac{1}{(\alpha + \beta) r} \right)$$
$$= \frac{1}{\alpha \beta (\alpha + \beta)} \sum_{r=1}^{\infty} \frac{1}{r^4} = \frac{\pi^4}{90} \frac{1}{\alpha \beta (\alpha + \beta)}. \tag{98}$$

The low-temperature regime for $(k, m, n)$ is defined by $\alpha \sim \beta$, unlike for $(1, 0, n)$, because $\alpha$ and $\beta$ enter $\log Z$ in the same way at the leading order. Using thermodynamic relations, we find

$$N = \frac{\pi^4}{90} \frac{2\alpha + \beta}{\alpha^2 \beta (\alpha + \beta)^2}, \qquad N_P = \frac{\pi^4}{90} \frac{\alpha + 2\beta}{\alpha \beta^2 (\alpha + \beta)^2}. \tag{99}$$

The condition $0 < \alpha \sim \beta \ll 1$ means that

$$1 \ll N \sim N_P. \tag{100}$$

Solving (99) for $\alpha, \beta$ and plugging in the result, we find that the entropy is given by

$$S_{(k,m,n)} = \frac{2^{7/4} \pi}{3^{5/4} 5^{1/4}} \left[ 2(N^2 - N N_P + N_P^2)^{3/2} - (N - 2N_P)(2N - N_P)(N_P + N) \right]^{1/4}. \tag{101}$$

In [22], superstrata were counted from the CFT side and it was found that, for $N \sim N_P \sim J$, where $J := J_0^3$ is the R-charge, the entropy is given by [22, eq. (4.77)]

$$S_{\text{CFT strata}} \propto \left[ J(N - J)(N_P - J) \right]^{1/4}. \tag{102}$$

By maximizing this with respect to $J$, it is straightforward to show that this reduces to (102), up to the overall coefficient (which is due to the fact that we are considering a subsector of all possible superstrata). The entropy (101) behaves for small and large $N_P$ as[12]

$$S \approx \begin{cases} \frac{2^{5/4} \pi}{3^{1/2} 5^{1/4}} N^{1/4} N_P^{1/2} & (N_P \ll N), \\ \frac{2^{5/4} \pi}{3^{1/2} 5^{1/4}} N^{1/2} N_P^{1/4} & (N \ll N_P). \end{cases} \tag{103}$$

---

[11]We could include $n = 0$ modes, which are 1/4-BPS, not 1/8-BPS as generic superstrata. However, this would not make any difference to the thermodynamic quantities such as the entropy.

[12]In (101) we have already assumed that $N \sim N_P$. So, $N_P \ll N$ and $N \ll N_P$ here are within the extent that we do not change the parametric scaling between them.

This is the same entropy growth found in [22] (see eq. (4.90) there), again, up to the overall coefficient. This is parametrically smaller than the D1-D5-P black hole entropy $S_{\text{BH}} \sim N^{1/2}N_P^{1/2}$.

The high-temperature regime, $0 < \beta \ll 1 \ll \alpha$, can be worked out almost the same way as for $(1,0,n)$. Because $p \ll 1$, the partition function (97) can be approximated as

$$\log Z \approx \frac{2p}{\beta}. \tag{104}$$

From this, we can derive

$$N = \frac{2p}{\beta}, \qquad N_P = \frac{2p}{\beta^2}, \qquad \text{therefore} \qquad p = \frac{N^2}{2N_P}, \qquad \beta = \frac{N}{N_P}. \tag{105}$$

$p \ll 1$ means that $N_P \gg N^2$. The entropy is

$$S \sim N\left(2 + \log \frac{2N_p}{N^2}\right). \tag{106}$$

This is parametrically smaller than the Cardy growth $S_{\text{CFT}} \sim \sqrt{NN_P}$. As mentioned at the end of section 4.2, the relevance of the high-temperature regime within AdS/CFT is unclear.

## Acknowledgments

We would like to thank Andrea Puhm, Robert Walker, and Nick Warner for useful discussions. DRM is supported by the ERC Starting Grant 679278 Emergent-BH and ERC Advanced Grant 787320 - QBH Structure. MS thanks the CEA Saclay for hospitality. The work of MS was supported in part by MEXT KAKENHI Grant Numbers 17H06357 and 17H06359.

## A   Symplectic Form Review

In this appendix, we will briefly review the formalism of the phase space symplectic form and its relation to the Poisson bracket and time evolution. For definitiveness, we follow the normalizations of [24].

The symplectic form is a two-form $\Omega$ on the even-dimensional phase space manifold $M$ of a physical system. For any one-form on this phase space, it can define a vector through:

$$V: \quad T^*M \to TM, \quad \alpha \mapsto V_\alpha; \quad i_{V_\alpha}\Omega = \alpha, \tag{107}$$

so that $i_{V_\alpha}\Omega(w) = \Omega(V_\alpha, w)$. For a function $f$ on the phase space, the natural associated vector is $V_f := V_{df}$. The Poisson bracket of two functions is then given by:

$$\{f, g\}_{\text{PB}} = -\Omega(V_f, V_g). \tag{108}$$

Given the Hamiltonian $H(x^A)$, the time evolution of a function $f$ is simply given by:

$$\frac{d}{dt}f = \{f, H\}_{\text{PB}}. \tag{109}$$

If we introduce coordinates $x^A$ on the phase space, then the symplectic form can be written as:

$$\Omega = \frac{1}{2}\omega_{AB}\delta x^A \wedge \delta x^B. \tag{110}$$

The components of the vector $V_\alpha$ associated to a one-form $\alpha = \alpha_A \delta x^A$ is given by:

$$V_\alpha^A = -\omega^{AB} \alpha_B, \tag{111}$$

where we used that $\omega_{AB}$ is antisymmetric, and $\omega^{AB}$ is the inverse matrix of $\omega_{AB}$. For a function $f(x^A)$, we have:

$$V_f = V_{df} = -\omega^{AB} \partial_B f. \tag{112}$$

Finally, the Poisson bracket is given by:

$$\{f, g\}_{\text{PB}} = \omega^{AB} \partial_A f \, \partial_B g. \tag{113}$$

In particular, it follows that:

$$\{x^A, x^B\}_{\text{PB}} = \omega^{AB}. \tag{114}$$

Time evolution can also be written in coordinate form as:

$$\frac{d}{dt} f = \{f, H\}_{\text{PB}} = \omega^{AB} \partial_A f \, \partial_B H. \tag{115}$$

As a simple example, the Lagrangian, Hamiltonian, and symplectic form for a simple, free particle in one dimension with position $q$, mass $m$, and momentum $p$ is given by:

$$L = \frac{1}{2} m \dot{q}^2, \qquad H = \frac{p^2}{2m}, \qquad \Omega = \delta p \wedge \delta q. \tag{116}$$

This implies that $\omega_{pq} = -\omega_{qp} = 1$ and so also $\omega^{qp} = +1$, which gives the canonical Poisson bracket:

$$\{q, p\}_{\text{PB}} = 1, \tag{117}$$

and the correct time evolution, for example:

$$\frac{dq}{dt} = \omega^{AB} \partial_A q \partial_B H = \omega^{qp} \partial_p H = \frac{p}{m}. \tag{118}$$

# B  Review of Rychkov's Consistency Condition for D1-D5

In this appendix, we briefly review the crucial steps of Rychkov's consistency condition arguments for the D1-D5 symplectic form[13] [9], which immediately leads to the correct symplectic form for the D1-D5 Lunin-Mathur supertube geometries, up to an overall constant.

## B.1  The consistency condition

For a general Hamiltonian system $(H, \Omega)$, we can restrict to a subsystem $\mathcal{M}$ which is invariant under Hamiltonian evolution, and define the restrictions of $H, \Omega$ on this subspace:

$$h := H|_{\mathcal{M}}, \qquad \omega := \Omega|_{\mathcal{M}}. \tag{119}$$

Then Rychkov's key theorem, the *consistency condition*, is the statement that on $\mathcal{M}$, the flows $(H, \Omega)$ and $(h, \omega)$ are equivalent.

In principle, to calculate the symplectic form in supergravity, one must consider the relevant solutions in the full, ten-dimensional supergravity action. However, this consistency

---

[13]Another work where the D1-D5 symplectic form was calculated explicitly in supergravity is [36].

condition immediately implies that, if all of the solutions we are interested in live in a subsector of this full supergravity, we can simply restrict ourselves to the (consistent) truncation of the ten-dimensional supergravity.

It is important to note that while Rychkov [9] uses this consistency condition for time-independent solutions, all that is really required for this consistency condition to hold is that the entire subspace $\mathcal{M}$ is invariant under the Hamiltonian evolution — the independent solutions need not be invariant.

Rychkov's consistency condition then immediately implies (in section 2) that we can simply restrict ourselves to the on-shell Hamiltonian of the superstrata. In section 3, this consistency condition implies that we can restrict ourselves to calculating the symplectic form directly in the three-dimensional supergravity theory where the $(1,0,n)$ superstrata live in, without having to resort to more complicated, higher dimensional theories.

## B.2 D1-D5 symplectic form, quantization, and counting

We will not review the D1-D5 Lunin-Mathur supertube geometries [4–6] in detail here, but only mention their most relevant features. (A succinct summary of the solutions in type IIB can be found in [9].) The D1-D5 geometries are completely smooth (in ten dimensions), horizonless, and are characterized by four arbitrary periodic functions $F_i(s)$ (with $i = 1, \cdots, 4$, and $F_i(s) \sim F_i(s+L)$) that determine a closed curve[14] in $\mathbb{R}^4$. The geometry further is dependent on the parameters $Q_5$ (the D5-brane charge) and $R_y$ (the radius of the $S^1$ in six dimensions). Note that the parameter period is $L = 2\pi Q_5/R_y$. The D1-brane charge $Q_1$ of the geometry is then given by:

$$Q_1 = \frac{Q_5}{L} \int_0^L |\vec{F}'(s)|^2 ds. \tag{120}$$

The degeneracy of the D1-D5 system with fixed charges $Q_1, Q_5$ is then given by the counting of the number of curves $\vec{F}$ that satisfy (120); classically, there are infinitely many such curves, but after quantization this becomes a well-posed question with a finite answer.

In particular, using units where $G_5 = \pi/4$ (see also section 3.2), the Hamiltonian of this system is simply the BPS energy, so:

$$H_{D1-D5} = Q_5 + Q_1 = Q_5 + \frac{Q_5}{L} \int_0^L |\vec{F}'(s)|^2 ds. \tag{121}$$

The D1-D5 geometries are time-independent, and also invariant under shifts of the parameter $F(s) \to F(s+c)$. It follows that the only possible allowed Hamiltonian evolution is:

$$\frac{d\vec{F}}{dt} = c\frac{d\vec{F}}{ds}, \quad \leftrightarrow \quad \vec{F}(s,t) = \vec{F}(s+ct,t). \tag{122}$$

This implies (using the prime to denote $s$-derivatives):

$$\frac{d\vec{F}}{dt} = \{\vec{F}, H\}_{\text{PB}} = \alpha F' = 2\left(\frac{Q_5}{L}\right)\omega^{FF'}F', \tag{123}$$

which immediately give the symplectic form[15]:

$$\Omega = 2\frac{Q_5}{\alpha L} \int ds\, \delta F'(s) \wedge \delta F(s), \tag{124}$$

---

[14]This curve should further be taken to be non-self intersecting, and should satisfy $|\vec{F}'(s)| \neq 0$ everywhere.

[15]Note that we are using a different normalization than Rychkov [9], so our $\alpha$ is different than his.

and the Poisson bracket:

$$\{F_i(s), F'_j(s')\}_{\text{PB}} = \frac{\alpha L}{2Q_5} \delta_{ij} \delta(s - s'). \tag{125}$$

The consistency condition, together with knowledge of the Hamiltonian of the solutions, has determined the symplectic form (124) up to a constant $\alpha$. To further determine this constant, an explicit computation in supergravity is needed; this explicit computation then shows [9]:

$$\alpha = \frac{2Q_5}{L} \pi \mu^2, \tag{126}$$

where $\mu = g_s/(R_y V_4^{1/2})$, using the string coupling $g_s$ and the volume $V_4$ of the $T^4$ which the D5 branes wrap.

We can also rewrite the symplectic form (124) and the Poisson bracket (125) in terms of individual oscillators. We can expand:

$$\vec{F}(s) = \mu \sum_{k=1}^{\infty} \frac{1}{\sqrt{2k}} \left( \vec{c}_k e^{i\frac{2\pi}{L}ks} + \vec{c}_k^{\dagger} e^{-i\frac{2\pi}{L}ks} \right). \tag{127}$$

Integrating the Poisson bracket then gives:

$$\{c_k^i, c_l^{\dagger j}\}_{\text{PB}} = +i \int_0^L ds \int_0^L ds' \frac{\sqrt{2k}}{\mu L} \frac{1}{\pi\mu\sqrt{2l}} e^{-i\frac{2\pi}{L}ks} e^{i\frac{2\pi}{L}ls'} \{F^i(s), F'^j(s')\}_{\text{PB}} \tag{128}$$

$$= i\delta^{ij} \int_0^L ds\, e^{i\frac{2\pi}{L}(l-k)s} \frac{\sqrt{2k}}{\sqrt{2l}} \frac{1}{\mu^2 \pi L} \left( \frac{\alpha L}{2Q_5} \right) \tag{129}$$

$$= i\delta^{ij} \delta_{kl} \frac{\alpha L}{2Q_5 \mu^2 \pi}. \tag{130}$$

With $\alpha$ as given in (126), we get the canonical Poisson bracket

$$\{c_k^i, c_l^{\dagger j}\}_{\text{PB}} = i\delta^{ij} \delta_{kl}, \tag{131}$$

as expected.

We can pass from the Poisson bracket (131) to the quantum commutator:[16]

$$[c_k^i, c_l^{\dagger j}] = \delta^{ij} \delta_{kl}. \tag{132}$$

The condition (120) can be rewritten as:

$$\sum_{k=1}^{\infty} k \langle c_k^{\dagger i} c_k^i \rangle = N_1 N_5, \tag{133}$$

where $N_1, N_5$ is the number of D1, D5 branes. Then, (133) corresponds to the $(N_1 N_5)$-th energy level of a CFT of 4 (since $i = 1, \cdots, 4$) chiral bosons ($c = 4$), with entropy (for large $N_1 N_5$):

$$S = 2\pi \sqrt{\frac{c}{6} N_1 N_5} = 2\pi \sqrt{\frac{2}{3} N_1 N_5}. \tag{134}$$

This corresponds to a finite fraction of the full D1-D5 entropy. If one additionally allows for curves $\vec{F}$ on the compact $T^4$, then the full D1-D5 entropy is reproduced by this counting [8,10].

---

[16]Here, one uses the map $\{f, g\}_{PB} \to -i[\hat{f}, \hat{g}]$; the normal ordering ambiguity in (121) allows us to choose the sign of this particular quantum commutator. See also [37].

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
