# Peer review of "Counting D1-D5-P Microstates in Supergravity"

_SciPost Physics, doi:SciPost Phys. 10, 018 (2021)_

## Round 1 · Referee Report · Suvrat Raju · 2020-12-19

Strengths
1. The authors obtain an elegant result by quantizing of a set of rather intricate solutions.
2. Paper is written concisely and clearly.
Weaknesses
None particularly.
Report
This is a very nice paper that describes how the superstrata geometries can be quantized directly in supergravity.
The space of classical solutions in a theory can be put in one-to-one correspondence with points in the theory's phase space. This is because, given any classical solution, one can evaluate the "position and momentum" at a point of time to obtain a point in phase space. Conversely, given a point in phase space, one can evolve it to obtain an entire classical solution. Therefore it is possible to quantize the "space of solutions" using the symplectic form on phase space. In the case at hand, the authors want to quantize only a subspace of solutions since we do not know all solutions in supergravity. But if one assumes that the symplectic form is closed on this subspace, then this is also believed to be a meaningful exercise.
The superstrata geometries that the authors wish to quantize in this paper are described in equation 2.1. These solutions depend on three integers, (k,m,n). The authors first write down a reduced Hamiltonian on the space of (1,0,n) superstrata in equation 2.16. They use this to guess a symplectic form in equation 2.19.
Of course, the symplectic form can also be directly obtained from the supergravity action. The authors perform a rather intricate computation in section 3 to check the symplectic form obtained in equation 2.19.
The authors also obtain a symplectic form for the (k,m,n) geometries in equation 2.33. They do not verify this through a supergravity computation since they "do not think this would be an interesting exercise." They claim that their guess must be correct due to the simple form of the reduced Hamiltonian.
The symplectic form obtained by the authors corresponds to a set of decoupled simple harmonic oscillators, and so the authors use this to "count" solutions in section 4.
Requested changes
I think this paper can be published largely as is. Two suggestions are as follows:
1) Perhaps the authors should introduce the (1,0,n) solutions before discussing them. At the moment, they appear without explanation early in the introduction and not all readers will be familiar with what the three numbers mean.
2) Perhaps the authors could add hep-th/0505079 to their list of references.
(in reply to Report 1 by Suvrat Raju on 2020-12-19)
We thank prof. Raju for his time in formulating this careful report and comments.
In v2 (as also mentioned on the v2 submission page), we have added reference [23] which is the paper hep-th/0505079 suggested in point 2) of his comments.
To address point 1), we have also added a sentence at the end of paragraph 4 in the Introduction (p2) to introduce the (k,m,n) notation for superstrata and also refer the reader to the section below where it is explained further.

---

## Round 1 · Referee Report · Samir Mathur · 2021-1-10

Strengths
Clearly written paper
Addresses an important problem in theoretical physics: the construction of microstates for black holes
Report
This is a well written paper which completes a step in the construction of superstrata geometries. It has been known since earlier work by de Boer that multigraviton states should contribute to the index of the D1D5P system. But these multigraviton states will have a nontrivial interaction between the gravitons as one is not in the regime where the number of gravitons is small. Remarkably, the nonlinear solutions for this situation can be constructed as classical supergravity solutions for the case the number of gravitons per mode is large; this allows the state to be approximated by a coherent state and thus attributed a classical solution. The resulting solutions are termed superstrata.
To actually count the space of quantum solutions of this type, one can quantize the space of classical solutions. This is what the authors manage to carry out, and they find a result in agreement with the number of multigraviton states expected from the index. The computation is a nontrivial one requiring an understanding of the quantization of supersymmetric phase space.
The authors note that the number of states obtained from this quantization is below the number given from the Bekenstein entropy; this is as expected since as noted in the index computation of de Boer, the black hole has other states that are not just made of collections of gravitons.
In summary this is a well written paper carrying out an nontrivial computation on an important topic; thus it should be published.

---

## Round 2 · List of Changes

v2: reference [23] added; sentence added at the end of paragraph 4 in introduction (p2) introducing superstrata (k,m,n) notation.

---

## Editorial Decision

published